# Transcriptomic Analysis of Radish (*Raphanus sativus* L.) Roots with *CLE41* Overexpression

**DOI:** 10.3390/plants11162163

**Published:** 2022-08-20

**Authors:** Ksenia Kuznetsova, Irina Dodueva, Maria Gancheva, Lyudmila Lutova

**Affiliations:** Department of Genetics and Biotechnology, Saint Petersburg State University, 199034 Saint Petersburg, Russia

**Keywords:** CLE41, *Raphanus sativus*, RNA-seq, differential gene expression, late embryogenesis, xylem specification, desiccation tolerance

## Abstract

The CLE41 peptide, like all other TRACHEARY ELEMENT DIFFERENTIATION INHIBITORY FACTOR (TDIF) family CLE peptides, promotes cell division in (pro-)cambium vascular meristem and prevents xylem differentiation. In this work, we analyzed the differential gene expression in the radish primary-growing *P35S:RsCLE41-1* roots using the RNA-seq. Our analysis of transcriptomic data revealed a total of 62 differentially expressed genes between transgenic radish roots overexpressing the *RsCLE41-1* gene and the glucuronidase (*GUS*) gene. For genes associated with late embryogenesis, response to abscisic acid and auxin-dependent xylem cell fate determination, an increase in the expression in *P35S:RsCLE41-1* roots was found. Among those downregulated, stress-associated genes prevailed. Moreover, several genes involved in xylem specification were also downregulated in the roots with *RsCLE41-1* overexpression. Unexpectedly, none of the well-known targets of TDIFs, such as *WOX4* and *WOX14*, were identified as DEGs in our experiment. Herein, we discuss a suggestion that the activation of pathways associated with desiccation resistance, which are more characteristic of late embryogenesis, in roots with *RsCLE41*-overexpression may be a consequence of water deficiency onset due to impaired vascular specification.

## 1. Introduction

The family of CLE (CLAVATA3/ENDOSPERM SURROUNDING REGION) peptide phytohormones includes small mobile peptides which act as short- and long-distance signals in the regulation of plant development, especially in stem cell homeostasis and cell fate determination. The most known function of CLE peptides is the regulation of meristem maintenance: for instance, CLAVATA3 restricts the stem cells’ population in the shoot apical meristem, and CLE40 fulfills a similar function in the root apical meristem [1].

The CLE41/CLE44 and CLE42 peptides, also known as TRACHEARY ELEMENT DIFFERENTIATION INHIBITORY FACTORs (TDIFs), promote cell division in (pro-) cambium vascular meristem and prevent the differentiation of xylem elements—especially vessels [2,3,4]. The TDIFs are ligands of the Leucine Rich Repeat-Receptor-Like Kinase (LRR-RLK) family protein, named PHLOEM INTERCALATED WITH XYLEM/TDIF RECEPTOR (PXY/TDR) [4,5]. In vascular bundles, TDIF-encoding *CLE41*, *CLE42* and *CLE44* genes are expressed in phloem, whereas the expression of PXY/TDR takes place in the (pro-)cambium [2,3,4,5]. TDIF binding to the PXY/TDR receptor independently regulates two processes. Firstly, TDIF interaction with PXY/TDR leads to the upregulation of *WUSCHEL RELATED HOMEOBOX 4* (*WOX4*) and *WOX14* genes. These genes then encode the transcription factors (TFs) which promote vascular cell division, thereby regulating (pro-)cambium proliferation [4,6]. Secondly, TDR/PXY, along with the Glycogen Synthase Kinase 3 (GSK3) family proteins, such as BRASSINOSTEROID INSENSITIVE 2 (BIN2), BIN2-LIKE1 (BIL1) and BIL2, negatively regulate the differentiation of vascular stem cells into xylem. The binding of TDIFs leads to the dissociation of the TDR/PXY-GSK3 complex which results in the phosphorylation and degradation of BES1 TF which promotes xylem differentiation [7]. In addition, GSK3 signaling regulates auxin response via the phosphorylation of the MONOPTEROS (MP) TF which is also required for vascular development [7,8].

The family of *Raphanus sativus CLE* genes (*RsCLEs*) and their role in the radish taproot development were previously described by [9]. According to the genomic sequence of *R. sativus* [10], the TDIF-encoding gene family in this plant species includes two closely related *RsCLE41* genes (LOC108857555 and LOC108857305, or *RsCLE41-1* and *RsCLE41-2*) and two *RsCLE42* genes (LOC108855008 and LOC108837535). Two *RsCLE41* genes and two *RsCLE42* genes demonstrate high similarity in their sequences. It was shown that the overexpression of *RsCLE41* genes, as well as synthetic CLE41 peptide treatment, have led to multiple apparent effects on secondary growing taproots and stems in mature radish plants. The plants acquired characteristics such as cambium proliferation in the stem and the root; the overgrowth of stem vascular bundles; the formation of the extra cambium-like foci in the xylem zone of the root stele; in addition to s suppressing of thick-walled lignified xylem development [9,11]. These data are consistent with the effects of *CLE41* overexpression or exogenous CLE41 treatment in other plant species [2,12,13] and could be explained by the well-known effects of *CLE41* on the *WOX4* and *WOX14* expression [6] and on BES1 TF stability [7].

At the same time, along with those already described, there may be other putative targets of TDIF peptides, and the transcriptomic analysis of plants overexpressing TDIF-encoding genes can serve as one of the methods for their identification. For instance, a transcriptomic study on the 5-week-old stem of *35S:CLE41 Arabidopsis* lines revealed the expression change in a number of genes involved in xylem cell differentiation or transcriptional regulation [13]. In the transcriptomic analysis of the 21 day-old *35S:CLE42 Arabidopsis* lines, numerous data explaining the negative effect of TDIF peptides on leaf senescence were obtained [14]. 

Here, we present a transcriptomic analysis of transgenic radish roots with *RsCLE41-1* (LOC108857555) overexpression. We identified a total of 62 differentially expressed genes (DEGs) in *35S:RsCLE41-1* transgenic radish roots (*CLE41-oe* from this point onward) compared with the control, the transgenic roots overexpressing glucuronidase (*GUS*) gene (*35S:GUS*, or *GUS-oe* from this point onward). Among the upregulated ones, genes which are normally expressed during late embryogenesis were most widely represented in *CLE41-oe* radish roots. As for the downregulated ones, stress-related genes as well as several genes involved in xylem development were shown to be differently expressed in our analysis. 

## 2. Results

### 2.1. RNA-Sequencing of Transgenic Radish Roots with RsCLE41 Overexpression and Those with GUS Overexpression

To compare the transcriptomes of radish roots with *RsCLE41-1* overexpression and mock-inoculated roots, we isolated the RNA of the transgenic roots of radish line 19 from the genetic collection of *R. sativus* [15] and subjected it to sequencing. Total RNA was extracted in triplicate and sequenced with an Illumina HiSeq2500 sequencer resulting in 60 million paired-end reads. After all adapter trimming and contamination removal steps, 55.2 million paired-end reads were used for differential expression estimation and all downstream analyses. After alignment with HISAT2 [16], approximately 82% of reads in each sample were uniquely mapped. Reads were counted by StringTie [17] and correlation analysis performed for DESeq normalized counts demonstrated high correlation between biological replicates [18]. After the analysis of the differential expression with the DESeq package and imposition of the 0.05 adjusted *p*-value and 2.0 log2 fold change cutoffs, 62 out of 70,111 analyzed genes were found to be differentially expressed, with 38 and 24 upregulated and being downregulated, respectively, in *CLE41-oe* roots in comparison to *GUS-oe* roots (Figure 1, Appendix A). 

We also performed gene ontology enrichment analysis with the GSEABase package using GO terms [19]. It was found that 52 and 8 “biological process” GO terms were overrepresented among the upregulated and downregulated genes, respectively (Figure 2). Based on our individual transcripts data, we constructed a gene network which reflects the relationships between genes under *RsCLE41-1* overexpression conditions (Figure 3). The qPCR expression analysis of 11 differentially expressed genes (DEGs) supported transcriptome analysis data (Figure 4).

### 2.2. Enriched Pathways in Genes Differentially Expressed between Roots with RsCLE41 Overexpression and Those with GUS Overexpression

Using pathway enrichment analysis with the GSEA package (*p*-value and odds ratio thresholds were 0.05 and 2, respectively) [20], we found 52 upregulated pathways and eight downregulated pathways in “biological process” GO terms (Figure 2). Among the upregulated ones, there were pathways associated with the response to the abiotic (GO:0032101) and biotic stimuli (GO:0002831); developmental processes (GO:0022611, GO:0010431, GO:0009933, GO:0021700, GO:0009888, GO:0010154 and GO:0009793), especially meristem development (GO:0010228 and GO:0090567); reproduction processes (GO:0032504, GO:0061458 and GO:0022414); signal transduction (GO:0007165), and various metabolic pathways (GO:0060255 and GO:0006631). Enriched upregulated pathways related to the response to biotic stimuli included the response to phytohormones (salicylic acid (GO:0009751) and ABA (GO:0009737)), defense response to fungus (GO:0050832) and bacterium (GO:0042742) and response to wounding (GO:0009611). Among the response to stimuli of a nonbiological nature, the enrichment of pathways associated with response to chitin (GO:0010200), nitrogen compound (GO:1901698), and water (GO:0009415) was found. As for the downregulated, the pathways associated with flavonoid biosynthesis (GO:0009963) and metabolism (GO:0009812) were enriched, as were those associated with aging (GO:0007568), response to starvation (GO:0009267), nutrient levels (GO:0031667), extracellular stimulus (GO:0031668), wounding (response to wounding) and the regulation of plant-type hypersensitive response (GO:0010363).

In ”cellular component” GO terms, there was a single upregulated pathway associated with the intrinsic component of a membrane (GO:0031224). As for downregulated genes, terms associated with “extracellular region” (GO:0005576), “plasmodesmata” (GO:0009506), “Golgi apparatus” (GO:0005794) and ”vacuole” (GO:0005773) were enriched. 

In ”molecular function” GO terms, there were two upregulated pathways: associated with sequence-specific DNA binding (GO:0043565); and DNA-binding transcription factor activity (GO:0003700). In these GO terms, there were also downregulated pathways associated with cysteine-type peptidase activity (GO:0008234), oxidoreductase activity, acting on paired donors, with the incorporation of or reduction in molecular oxygen (GO:0016705) and protein binding (GO:0005515).

### 2.3. Individual Transcripts Differentially Expressed between Roots with RsCLE41 Overexpression and Those with GUS Overexpression

In our experiment, 62 genes were identified as differentially expressed between *CLE41-oe* and *GUS-oe* radish roots: among them, 38 genes were significantly upregulated and 24 genes were downregulated in *CLE41-oe* roots. According to ”biological function” GO terms, these genes belong to 52 upregulated pathways and 8 downregulated pathways. 

The log2 fold change for genes with significantly changed expression levels ranged from 8.24 for the upregulated genes to −7.54 for the downregulated genes. For example, the *RsCLE41-1* expression level was significantly increased (log2 fold change was 6.048) (Appendix A). Based on our individual transcripts data, ”traditional” TDIF targets such as *WOX4* and *WOX14* were not identified as DEGs in our experiments: their expression levels were not statistically significantly changed in *CLE41-oe* roots compared to *GUS-oe* roots.

Among DEGs in *CLE41-oe* roots, there were genes essential for the overall development of plants. Among upregulated genes, there were radish homologs of *Arabidopsis* genes encoding calmodulin-like protein CML30, which localized in mitochondria and probably integrated into the cellular calcium/calmodulin signaling pathways [21]; arabinogalactan protein AGP1, which is a component of cell walls implicated in an extremely diverse aspects of plant growth and development [22]; and LONG VEGETATIVE PHASE 1 NAC domain TF (LOV1/NAP35), which participates in overall plant development, cell wall biosynthesis and flowering [23]. At the same time, the radish homolog of *Arabidopsis PROFILIN1 (PRF1*) encoding a conserved actin monomer-binding protein which regulates actin filament dynamics during axial cell expansion [24] was downregulated in our experiment. 

In our experiment, among the upregulated genes, we identified those belonging to the conservative groups that perform a broad set of functions in the overall development of plants. Nevertheless, the genes identified in our study were only executed in a certain development program. For example, the radish homologs of genes encoding mitochondrial uncoupling protein 4 (UCP4), a participant in mitochondrial reactive oxygen system (ROS) homeostasis and ARGONAUTE9 (AGO9), a target of the small RNAs, can both be mainly involved in gametophyte development and the regulation of fertility as well as the corresponding Arabidopsis genes [25,26].

Among genes which were upregulated in *CLE41-oe* roots, there were genes related to late embryogenesis and auxin-dependent xylem cell fate determination. At the same time, only among those downregulated in *CLE41-oe* roots were there several genes responsive for lignin biosynthesis. There were also numerous DEGs involved in biotic and abiotic stress response: among the upregulated stress-responsive genes, there were those involved in the response to ABA, mainly encoding TFs; among the downregulated genes, there were regulators of antimicrobial activity.

#### 2.3.1. Genes Involved in Late Embryogenesis: Strongly Upregulated

The most numerous functional group of genes upregulated in *RsCLE41-1* overexpressing roots consisted of the genes related to late embryogenesis. 

For example, five genes, which were upregulated in *CLE41-oe* roots at the highest level, encoded late embryogenesis abundant (LEA) proteins. The proteins of the LEA group are small and hydrophilic, mostly intrinsically disordered, and have exceedingly diverse subcellular localization [27,28]. These proteins play a major role in providing the desiccation tolerance, mainly during seed dormancy [29]. According to their primary sequences, LEA proteins can be grouped into eight families: dehydrin, LEA_1, LEA_2, LEA_3, LEA_4, LEA_5, LEA_6 and seed maturation protein (SMP) [27,30]. Among *LEA* genes which were upregulated in our experiment, there was radish *DC8* and *DC8-like* genes which are homologs of *A. thaliana ECP63* (LEA_4 subgroup), two *P8B6* genes which are homologs of *Arabidopsis EM6* (LEA_5 subgroup) as well as homologs of two *Arabidopsis* genes for two dehydrin subgroup proteins: *Xero1* and unnamed dehydrin *At1g54410.1*.

Several other genes upregulated in *CLE41-oe* radish roots are also related to late embryogenesis. Among them were two genes encoding seed storage proteins—radish homologs of *Arabidopsis* genes for vicilin-like seed storage protein (At2g28490) and oleosin 2 (*OLE2*). 

The gene encoding aquaporin TIP3-1 whose radish homologs were also upregulated in *CLE41-oe* roots also acts predominantly in plant embryogenesis. Aquaporins are integral membrane proteins which make pores that selectively pass water molecules through, enabling them to enter and leave the cell. At the same time, aquaporins prevent the flow of ions and other soluble substances from entering the cell. In *Arabidopsis*, *TIP3;1* and *TIP3;2* genes are specifically expressed during seed maturation, and their encoding proteins were shown to be localized in a seed protein storage vacuole membrane and to participate in the seed longevity [31].

Among those upregulated in *CLE41-oe* roots, there were also several enzyme-encoding genes whose homologs in *Arabidopsis* are predominantly expressed during late embryogenesis and seed development: genes for seed-specific peroxygenase 2 (*PXG2*), 1-cysteine peroxiredoxin 1 (*PER1*), large subunit GTPase 1 (*LSG1-1*) and the dehydrogenase of glucose and ribitol (At3g05260). 

Some of them encode antioxidant enzymes essential for plant defense against ROS substances. For example, PXG2 is a heme-containing enzyme with an EF-hand calcium binding motif which catalyzes the direct transfer of one oxygen atom from a hydroperoxide to a substrate to be oxidized [32]. Peroxiredoxins, including PER1, employ a thiol-based catalytic mechanism to eliminate ROS as peroxide reductases [33]. These genes are proposed to have an important function during seed development accompanied by a dramatic increase in the ROS level at seed desiccation. The dehydrogenase of glucose and ribitol, which is encoded by At3g05260, also has an oxidoreductase activity and expresses in the course of seed development.

#### 2.3.2. Genes Governing Xylem Formation: Upregulation of Genes Involved in Early Stages of Xylem Specification and Downregulation of Genes Participating in Late Stages of Vessel Development

Xylem formation is a key developmental process. It is operated by TDIFs [2,7,8] and includes early events of cambium cell fate determination to the xylem way and later events of vessel maturation, namely secondary cell wall production and cell death [34].

It is well known that CLE41 negatively affects the differentiation of xylem elements by interacting with its TDR/PXY receptor and its GSK3 family proteins. The GSK3s regulate the degradation of brassinosteroid-responsive BES1 TF which promotes xylem differentiation as well as the phosphorylation of the auxin-responsive MP TF which is also required for the vascular development [7,8].

In our transcriptome experiment, we revealed the upregulation of several genes involved in the auxin-dependent xylem cell fate determination, and the downregulation of several genes, including those related to lignin biosynthesis which participate in late events of vessel formation. In particular, among the upregulated in *CLE41-oe* radish roots, there was the gene-encoding homeodomain leucine zipper (HD-ZIPIII) TF PHABULOSA/ARABIDOPSIS THALIANA HOMEOBOX PROTEIN 14 (PHB/ATHB-14), as well as four radish homologs of *Arabidopsis* gene *LITTLE ZIPPER 4* (*ZPR4*).

HD-ZIPIII TFs are known to be the regulators of procambium development and metaxylem cell identity in the *Arabidopsis* root. The expression of genes encoding HD-ZIPIIIs can be promoted by high auxin levels [35]. In their turn, HD-ZIPIII TFs affect the auxin level governing the genes encoding core auxin response molecules [35,36,37]. In particular, PHB directly binds to the promoters of both MONOPTEROS/AUXIN RESPONSE FACTOR5 (MP/ARF5) and INDOLE ACETIC ACID 20 (IAA20), encoding the key antagonistic regulators of vascular development [37]. HD-ZIPIIIs including PHB also regulate the embryo patterning and the development of shoot apical meristem and its derivatives via auxin transport and signaling [36].

Small ZIP proteins are reported to be the main antagonists of HD-ZIPIII TFs [38,39,40]. For instance, ZPR3 and ZPR4 were shown to be competitive inhibitors of HD-ZIPIIIs and can directly interact with PHB and other HD-ZIPIIIs, generating nonfunctional heterodimers which negatively regulate the HD-ZIPIII activity in the SAM. At the same time, PHB can positively regulate *ZPR* genes expression constituting a feedback regulatory loop [39]. Thus, the upregulation of both *RsPHB* and *RsZPR4* in *CLE41-oe* roots suggests that they are the components of the same pathway.

Two more genes upregulated in *CLE41-oe* roots are engaged in the auxin-dependent morphogenetic processes, including vascular system development. 

One of these genes encodes a controller of the plant sterol modification pathway, sterol 4α-methyl oxidase1 (SMO1). SMO1 and SMO2 enzymes participate in the removal of the first and second methyl groups at the C-4 position of sterols, respectively. The *smo1-1 smo1-2* embryos exhibited severe defects such as impaired venation, associated with the enhanced and ectopic expression of *PIN1*, *PIN7* and *AXR1* genes regulating auxin transport and response [41]. 

The *LSG1* gene which also was strongly upregulated in *CLE41-oe* roots encodes a large subunit GTPase involved in the maturation of the ribosome 60S subunit. This protein is conserved in eukaryotes and is essential for cell viability in yeast. In *Arabidopsis*, the loss-of-function mutant of *LSG1* named *dig6* (*drought inhibited growth of lateral roots*) has multiple auxin-related abnormalities, namely shorter roots, reduced lateral root number as well as disrupted venation. These effects may probably evolve due to the altered auxin transport and auxin response in *LSG1*-deficient plants, thus hypothesizing the *LSG1* role both in ribosome biogenesis and auxin homeostasis [42].

At the same time, among those downregulated in *CLE41-oe* roots, there were several genes whose homologs in *Arabidopsis* are related to the biosynthesis of lignin and late stages of vessel specification, specifically genes encoding pirin 2, trans-cinnamate-4-hydroxylase-like, and two genes for glycine-rich proteins, namely DOT1 and DOT1-like.

The deposition of lignin polymers to the cell wall is a crucial factor for the differentiation and function of xylem vessels [34], and CLE41 can act antagonistically during this process [2,7]. 

The trans-cinnamate-4-hydroxylase (C4H) is a key enzyme of the phenylpropanoid pathway synthesizing monolignols—the building blocks of lignin polymers [43].

Pirin 2 (PRN2) belongs to a rather poorly described cupin domain-containing family of proteins that are conserved between prokaryotes and eukaryotes and function as transcriptional co-regulators. In *Arabidopsis*, PRN2 protein is specifically localized in the cells near the vessel elements and modulates lignin chemistry in the secondary cell walls of the neighboring vessels, suggesting a role for PRN2 in the noncell-autonomous cell wall lignification in xylem specification [44]. PRN2 suppresses the expression of lignin-biosynthetic genes and causes a decrease in syringyl (S)-type lignin biosynthesis [44]. In addition, PRN2 regulates the susceptibility to vascular colonizing bacteria, e.g., *Ralstonia solanacearum* [45].

Glycine-rich protein with unknown function DOT1 also seems to be involved in leaf vascular patterning, because *defectively organized tributaries 1 (dot1*) mutants have an aberrant venation pattern and defective vessels in the leaves [46].

Thus, the overexpression of *RsCLE41-1* in our experiments, according to the gene expression profile, enhanced the early events of xylem cell specification and led to the suppression of their final differentiation into xylem elements. This assumption was proven by our previously described histological observation on the decrease in lignified xylem element number in *R. sativus* roots with *CLE41-oe* [9,10,11].

#### 2.3.3. Stress-Related Genes: Upregulation of TF-Encoding Genes Involved in the Response to ABA, Ethylene and Jasmonates and Downregulation of Genes Acting in Disease Resistance

In our data, genes known to be regulators of plant response to biotic and abiotic stresses have been represented in both upregulated and downregulated groups. For example, *AtSAP18*, an orthologue of human *SAP18*, was the most upregulated in our experiment. This gene is a component of histone deacetylase complex conserved among eukaryotes which mediates transcriptional repression in the regulation of salt stress in *Arabidopsis* [47]. At the same time, the gene encoding defensin-like (DEFL) protein 206 was downregulated to the greatest extent in *CLE41-oe* roots. According to the literature data, DEFL206 as well as other plant defensins and DEFLs are involved in the impairing microbial activity and in the mediation of abiotic stress [48].

Most stress-related genes upregulated in *CLE41-oe* roots are involved in response to “stress-related” hormones as ABA, ethylene and jasmonates (JA) and were shown to encode the TFs of three families. 

Among them were three genes encoding the TFs of the WRKY family: two homologs of *Arabidopsis* WRKY18 and one homolog of WRKY40. In *Arabidopsis*, WRKY40 and WRKY18 are interacting TFs which play opposite roles in the response to biotic and abiotic stresses. It was shown that WRKY18, together with another partner WRKY60, acts as a weak transcriptional activator, while WRKY40 possesses a characteristic of a transcriptional repressor [49,50]. Both WRKY40 and WRKY18 (and also WRKY60) are targets of chloroplast-localized receptor of ABA, the magnesium-protoporphyrin IX chelatase H subunit (CHLH/ABAR): cytosolic C terminus of CHLH interacts with these WRKYs [51].

Another gene which was upregulated in our experiment is a radish homolog of *Arabidopsis* gene encoding JASMONATE-ASSOCIATED MYC2-LIKE1 (JAM1) TF. The product of JAM1, also called ABA-INDUCIBLE BHLH-TYPE TRANSCRIPTION FACTOR (AIB), positively regulates ABA response, in its turn, the expression of *JAM1/AIB* was transitory induced by ABA [52]. Another function of JAM1 is a negative regulation of JA signaling: JAM1 shows a high similarity to MYC2 TF, a major regulator of JA response, which can competitively bind to the target sequence of MYC2 acting as a transcriptional repressor [53].

The radish homolog of a gene encoding another transcriptional repressor, Ethylene Response Factor 11 (ERF11), which was upregulated in *CLE41-oe* roots, can integrate ABA and ethylene responses in Arabidopsis: AtERF11 was shown to be an ABA-responsive gene and a central regulator of the ABA-dependent decrease in ethylene biosynthesis [54]. In addition, *Arabidopsis* ERF11 positively regulates plant immunity to certain pathogens [55].

Thus, most genes upregulated in *CLE41-oe* roots are ABA-responsive TFs. The only two stress-related genes upregulated in our experiment and not directly related to the phytohormonal system are the genes encoding proteins conserved among eukaryotes. One of them is SAP18 (see above) and another is a radish homolog of gene encoding CCR4-CAF1, carbon catabolite repressor 4-CCR4 associated factor 1, the major enzyme complex that catalyzes mRNA deadenylation and degradation. It was shown that *AtCAF* genes are responsive for mechanical wounding and abiotic stresses and play a role in the defense response [56].

At the same time, among the genes downregulated in *CLE41-oe* roots, there were numerous genes involved, mainly in disease resistance. Our results show that 14 out of 23 downregulated genes can be classified as stress related. 

Among them, there are regulators of response to pathogens such as radish defensin-like protein 206 (RsDEFL206), neutral/alkaline non-lysosomal ceramidase, and three radish homologs of Arabidopsis Hypersensitive Induced Response protein 1 (HIR1). 

Plant defensins are small cysteine-rich peptides with antimicrobial activity [48]; in addition, each plant species contains numerous defensin-like (DEFL) peptides: for instance, 317 of DEFL genes were identified in Arabidopsis [57]. The *RsDEFL206* strongly downregulated in *CLE41-oe* roots is a homolog of Arabidopsis DEFL-encoding gene *At3g59930.* In *Arabidopsis*, *At3g59930* is predominantly expressed in the roots and is involved in resistance to parasitic nematodes [58].

Two radish isoforms of cysteine protease RDL5 have also demonstrated a notable reduction in the expression in *CLE41-oe* roots. *RDL5* of *Arabidopsis* is a member of intracellular NOD-like receptor family, and the R gene involved in disease resistance, which possesses leucine-rich repeat in its structure [59,60]. This gene shows resemblance to the *Arabidopsis thaliana* Papain family cysteine protease (PCP), AT4G11320, which is required for the regulation of photosynthetic gene expression, acclimation to high light stress [61] and programmed cell death [62]. It also takes part in plant development, senescence and defense responses [63]. In *Arabidopsis*, PCPs play a role in the programmed cell death of tracheary elements, tapetum and suspensor [64] as such, this gene can also act in the late stage of vessel specification.

In plants, membrane-localized HIR proteins, members of the conserved for eukaryotes proliferation, ion and death (PID) superfamily, have been shown to participate in the hypersensitive response lesions in leaves, a primary reaction to pathogen attacks [65]. The levels of HIR proteins were shown to be significantly induced by microbe-associated molecular patterns and bacterial effector proteins and correlated with localized host cell deaths and defense responses in different plant species [66,67,68,69]. Moreover, the overexpression of *HIR* genes caused enhanced pathogen resistance [66], but the mechanism of *HIR* gene involvement in plant immunity is still not clear.

In *Arabidopsis*, neutral/alkaline non-lysosomal ceramidase NCER affects sphingolipid homeostasis, plays a role in response to oxidative stress as well as participates in disease resistance and salt tolerance [70].

Thus, the overexpression of *RsCLE41-1* in our experiment seems to impair the disease resistance and plant cell death mechanisms and activate the ABA response.

### 2.4. Verification of Differential Expression Analysis Data with qPCR

We also performed the qPCR expression analysis for some genes that showed an increase or decrease in their expression levels in the *CLE41-oe* roots according to transcriptome analysis. 

Seven upregulated genes (genes for embryonic protein DC-8 (*RsDC8*), namely late seed maturation protein P8B6 (*RsP8B6*), dehydrin Xero 1-like (*RsXERO1*), oleosin 2 (*RsOLE2*), cyclinU2-2 (*RsCycU2*), MYC2-like TF (*RsMYC2*) and WRKY40 TF (*RsWRKY40*)); together with four downregulated genes (genes for pirin-like protein (*RsPIR2*), SRC2-like protein (*RsSRC2*), glycine-rich protein DOT1 (*RsDOT1*) and cysteine protease RDL5 (*RsRDL5*)) were selected for the qPCR analysis (Figure 1, Appendix A). In addition, we performed the qPCR analysis of the expression of the three genes involved in response to TDIFs, namely *RsPXY, RsWOX4* and *RsWOX14*. The expression levels of these genes, as in our transcriptomic experiment, did not show significant differences between the *GUS-oe* and *CLE41-oe* roots (Figure 4).

In general, the obtained results of qPCR confirmed the transcriptome data: genes that were identified as upregulated and downregulated in the transcriptome experiment demonstrated a significant increase or decrease in the expression levels in qPCR analysis, respectively (Figure 4).

## 3. Discussion

Thus, in our transcriptomic analysis of the roots with *CLE41-oe*, we obtained both expected and unexpected results. 

The most striking result to emerge from the data is that we detected no previously identified targets among the genes that change the expression levels in *CLE41-oe* roots. There was also no overlap with previously obtained data on the transcriptomes of leaves and stem vascular bundles with an overexpression of *TDIF* genes [13,14]. This surprising aspect of our data could be explained by the assumption that signaling pathways in the different parts of the plant can be aimed at different targets. Our transcriptomic data reflect changes in the gene expression in young *CLE41-oe* transgenic roots during the course of primary growth. Anatomically, these roots also did not show any significant differences from the roots with the overexpression of *GUS* (Figure 5).

Among the genes upregulated in *CLE41-oe* roots, there were genes, in particular *PHB* and *ZPR4*, involved in the early events of xylem specification control. At the same time, the downregulation of genes probably governing the vessel differentiation was detected in *35S:RsCLE41-1* roots. This group included genes participating in lignin biosynthesis (*C4H, PRN2*), the programmed cell death of tracheary elements (*RDL5*) and plant organs’ vascularization (*DOT1*). Thus, in our transcriptomic study, the data on gene expression change partly correspond to previous data regarding the CLE41-dependent suppression of xylem differentiation [2,9,11].

At the same time, it is well known that the central master regulators of the structure and composition of the vessel cell walls are genes encoding so-called VNSs (for VND, NST/SND and SMB-related protein) NAC domain TFs [34,71]. None of the genes belonging to this group were identified as differentially expressed between *CLE41-oe* and *GUS-oe* roots in our transcriptome experiment (Appendix A). On the other hand, among the “vessel-related” genes downregulated in *CLE41-oe* radish roots, only *C4H* gene belongs to the indirect targets of VND7 TF (Figure 3), but nothing is known about the transcriptional regulation of *PRN2, RDL5* and *DOT1* genes. Thus, we can speculate that there can be hypothetical VNS-independent minor pathways for regulating vascular development, which includes *PRN2, RDL5* and *DOT1* genes.

In addition, genes that act in other developmental programs were also identified as DEGs in *CLE41-oe* roots. This group is characterized by the wide representation of genes involved in late embryogenesis, and the participants of ABA response were among the upregulated genes. At the same time, there may be relationships between these groups of genes. For instance, ABA is known to be a major regulator of late embryogenesis and seed development, so most of the genes involved in these processes are ABA responsive. Indeed, on the one hand, among the aforementioned genes related to late embryogenesis, the ABA-dependent regulation of expression was documented for Arabidopsis *LEA* [72] and *TIP3* [31] genes, as well as for the genes involved in the storage process in seeds [73]. On the other hand, *PER1* in *Arabidopsis* seeds eliminates ROS to suppress ABA catabolism and GA biosynthesis [74]. The component of the histone-deacetylase complex SAP18 protein together with histone-deacetylase HDA1 can associate with ERF TFs, thus creating a hormone-sensitive multimeric repressor complex under conditions of environmental stress [47], so ABA-responsive *ERF11* and *SAP18* genes—which were upregulated in *CLE41-oe* roots—can also act in the same pathway.

Thus, in the *CLE41-oe* primary grown roots, we observed an increase in the expression of genes involved in the ABA response, as well as ABA-regulated genes that mainly act in late embryogenesis and regulate desiccation tolerance (*LEA* genes, *PER1, PXG2, TIP3*, etc.). A logical explanation for this effect may be the assumption that the activation of this pathway may be a consequence of the defects in vascular lignification and xylem specification in *CLE41-oe* roots. 

At the early stage of root development analyzed herein, we did not visually observe disturbances in the xylem development (Figure 5), but changes in the conductivity of water through such vessels may have already begun. This, in turn, can lead to the activation of the response to ABA and an upregulation of ABA-responsive genes which participate in the desiccation resistance, which normally are mainly triggered during late embryogenesis. Our assumption is schematically shown in Figure 6.

## 4. Materials and Methods

### 4.1. Construction of Vector

PCR fragments amplified on *R. sativus* DNA with primers for the full-length CDS of *RsCLE41-1* were cloned to pDONR221 vector (Ghent, Belgium) using the BP Clonase enzyme (Invitrogen, Waltham, MA, USA) and after that transferred to pB7WG2D vector (Ghent, Belgium) for overexpression using the LR Clonase enzyme (Invitrogen, Waltham, MA, USA). Constructs were introduced into *Agrobacterium rhizogenes* Arqua strain via electroporation using Eppendorf Eporator^®^ (Eppendorf, Hamburg, Germany).

### 4.2. Plant Growth, Transformation and Sample Collection

*R. sativus* inbred line 19 from the Saint Petersburg State University genetic collection [15] was used in the analysis. This line is approximately the 40th inbred generation and originated from the Saxa cultivar (European group of radish varieties). *R. sativus* seeds were surface-sterilized by 1:1 *v/v* mix of 90% ethanol and 30% hydrogen peroxide for 7 min, germinated on solid Murashige–Skoog (MS) medium [75] for 7 days at 21 °C at 16 h photoperiod.

The 7-day-old radish seedlings were subjected to inoculation with *A. rhizogenes* Arqua strain bearing *P35S:RsCLE41-1* (*CLE41-oe*) transgenic construction. *A. rhizogenes* Arqua strain bearing *P35S:GUS* (*GUS-oe*) was used in the case of control treatment. Infected plants were co-cultivated with bacteria on MS medium for 3 days, then transferred to MS medium with cefotaxime (500 mg/L) to kill the bacteria. The roots that grew immediately after inoculation with *A. rhizogenes* were cut off because they were non-transgenic or chimeric. After 7 days of growing, transgenic roots were selected by detecting the luminescence of the green fluorescent protein (GFP) reporter protein under a blue light. Three biological replicates for roots with *GUS-oe* and three biological replicates for roots with *RsCLE41-oe* were taken. Plant material was divided into two parts, the first for transcriptomic analysis and the second for quantitative PCR (qPCR) analysis.

### 4.3. RNA Isolation, Library Preparation and Sequencing

For RNA-seq, the total RNA from the transgenic roots of radish was isolated according to the modified phenol–chloroform method [76]. For both *CLE41-oe* and *GUS-oe* genotypes, three biological samples were used. The synthesis of cDNA was performed using the Mint-2 kit (Evrogen, Moscow, Russia). 

Library preparation and sequencing was performed by the Saint Petersburg State University Research Park, Centre for Molecular and Cell Technologies (Saint Petersburg, Russia). Illumina libraries were made from polyA RNA with NEBNext^®^ Ultra™ DNA Library Prep Kit (New England Biolabs, Hitchin, UK) according to the manual. Double barcoding was performed using the NEBNext^®^ Ultra™ DNA Index Prep Kit for Illumina and the NEBNext^®^ Multiplex Oligos^®^ Illumina^®^ (Dual Index Primers Set 1). The sequencing of libraries was performed on an Illumina HiSeq2500 sequencer with 50 bp read length.

### 4.4. Bioinformatics Processing of Sequencing Results

Raw fastq reads purification from mitochondrial, plastid and ribosomal radish DNA was carried out using the bbduk tool from bbtools suite (v. 38.96) (https://jgi.doe.gov/data-and-tools/bbtools/ accessed on 9 June 2022). Quality filtered and decontaminated reads were trimmed with Trimmomatic (v. 0.40) [77] with “HEADCROP:15 CROP:95” options. Read quality control was performed with MultiQC (v.1.12) [78]. Filtered reads were aligned on the sequences of *R. sativus* reference genome Rs1.0 (https://www.ncbi.nlm.nih.gov/assembly/GCA_000801105.2 (accessed on 9 June 2022)) with HISAT2 v. 2.01.2 [16]. Reads then were counted with Stringtie [17] with the use of the reference genome mentioned before and without de novo assembled transcripts. The differential expression between samples with *RsCLE41-oe* and control (*GUS-oe*) samples was analyzed with the DESeq2 package (v. 1.28.1) for R (v. 3.6.2) [18]. Genes with *s*-value < 0.05 and logFoldChange > 1 were considered differentially expressed. The GSEABase v.1.50 [20] R package was used for GO gene enrichment analysis. 

### 4.5. qPCR Expression Analysis

The data on the differential expression obtained by RNA-seq were verified by the qPCR of selected genes. For qPCR, total RNA was extracted by Purezol reagent (Bio-Rad, Hercules, CA, USA) according to the manufacturer’s instructions, purified with chloroform and precipitated with isopropanol. The RNA pellet was washed three times with 80% ethanol, dried under air flow in a laminar box and dissolved in sterile deionized water. RNA was treated with DNase I of the Rapid Out DNA Removal Kit (Thermo Fisher Scientific, Waltham, MA, USA) for DNA removal. RNA concentration was measured by the NanoDrop 2000 UV spectrophotometer (Thermo Fisher Scientific) at 260 nm. 

For reverse transcription, 500 ng of RNA was used in all samples. The RNA reverse transcription was performed using Revert Aid Reverse Transcriptase kit (Thermo Fisher Scientific) with oligo-dT18 primer according to the manufacturer’s instructions. To check the DNase treatment efficacy, the qRT-PCR analysis of the control samples without the reverse transcriptase was performed. 

The qRT-PCR experiments were performed on a CFX96 real-time PCR detection system with C1000 thermal cycler (BioRad, Hercules, California, USA) and an Eva Green intercalating dye (Syntol, Moscow, Russia) was used for detection. Primers for qRT-PCR (Appendix A) designed using the Primer3 Select online software [79] to amplify 150–250 bp fragments were synthesized by Evrogen (Moscow, Russia). The specificity of PCR amplification was confirmed based on the melting curve (55–95 °C). All reactions were performed in technical triplicate and averaged. 

Cycle threshold values were obtained with the accompanying CFX manager software, and data were analyzed by 2^−ΔΔCt^ method [80]. The relative expression was normalized against constitutively expressed *R. sativus* glyceraldehyde3-phosphate dehydrogenase (*RsGAPDH*) and polyubiquitin-like (*RsUBQ*) genes [81]. Experiments were repeated three times with independent biological samples, whose results were then averaged. 

## Figures and Tables

**Figure 1 plants-11-02163-f001:**
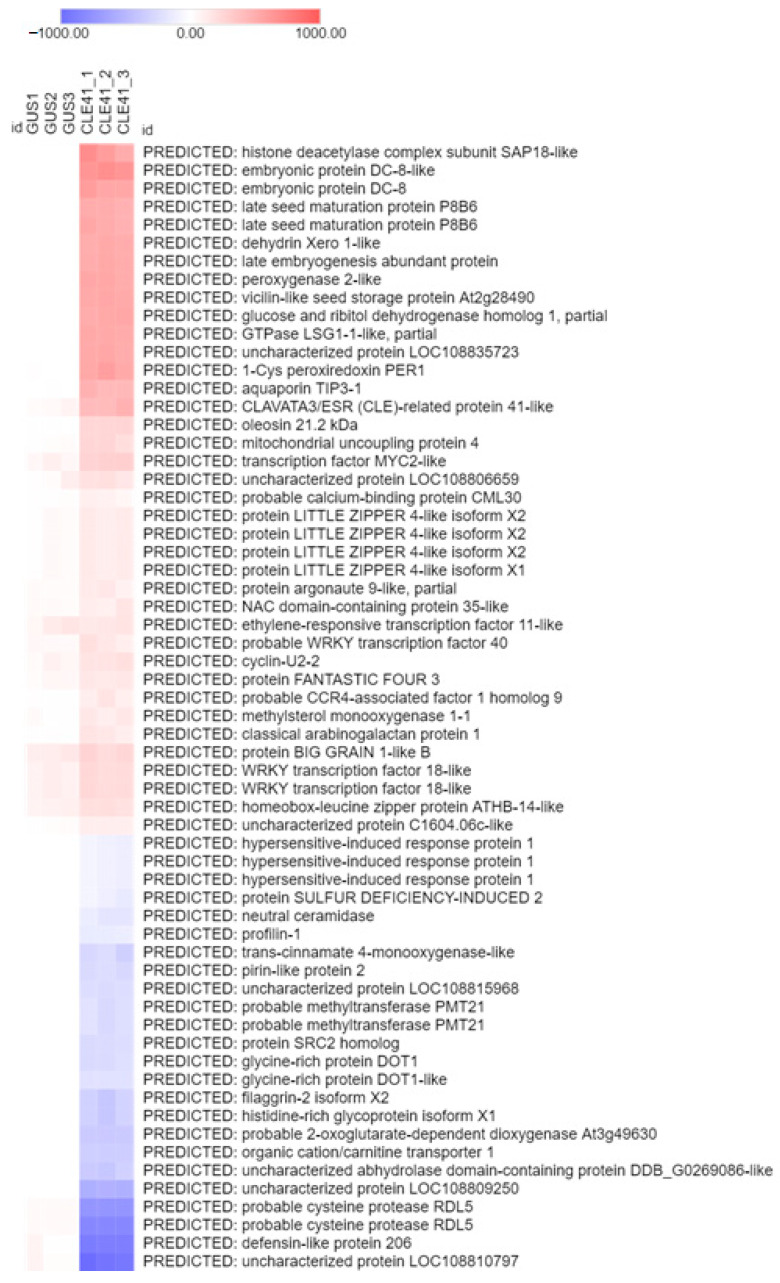
Heatmap representing expression levels of genes which were differentially expressed in the *35S:RsCLE41-1* roots in comparison to the *35S:GUS* roots of radish.

**Figure 2 plants-11-02163-f002:**
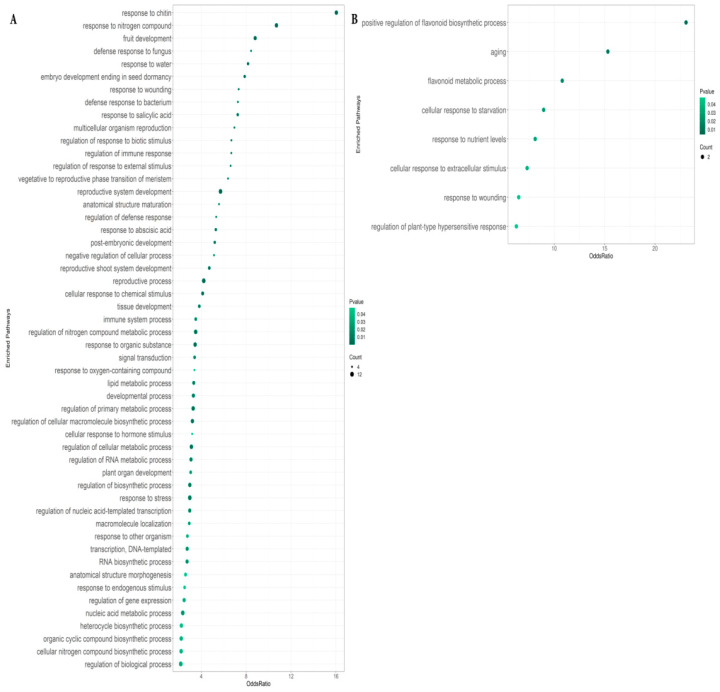
Overrepresented “biological process” GO pathways in genes upregulated (**A**) and downregulated (**B**) in the *35S:RsCLE41-1* roots in comparison to *35S:GUS* roots of radish. Count is the number of DEGs included in the respective pathway.

**Figure 3 plants-11-02163-f003:**
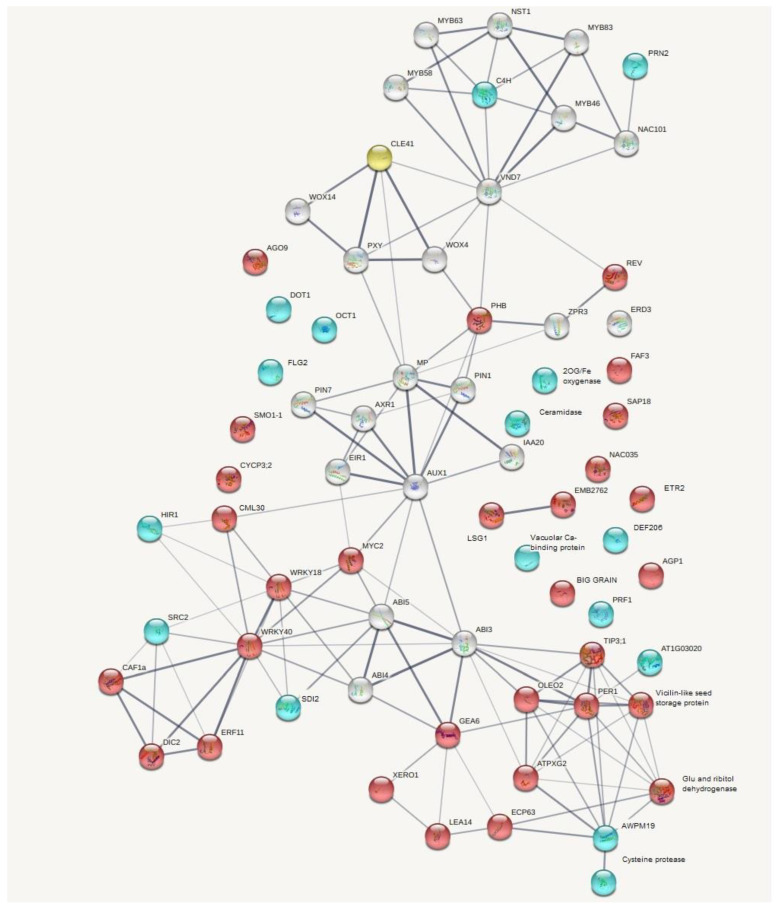
Gene network made by String (https://string-db.org/ (accessed on 2 August 2022)) representing individual transcripts’ data under *RsCLE41-1* overexpression conditions (*CLE41-oe*). Red color indicates genes upregulated under *CLE41-oe*; blue indicates those downregulated under *CLE41-oe*; white is for other participants of the *CLE41-oe* mediated network. *RsCLE41-1* is marked in yellow. Line thickness indicates the strength of data support.

**Figure 4 plants-11-02163-f004:**
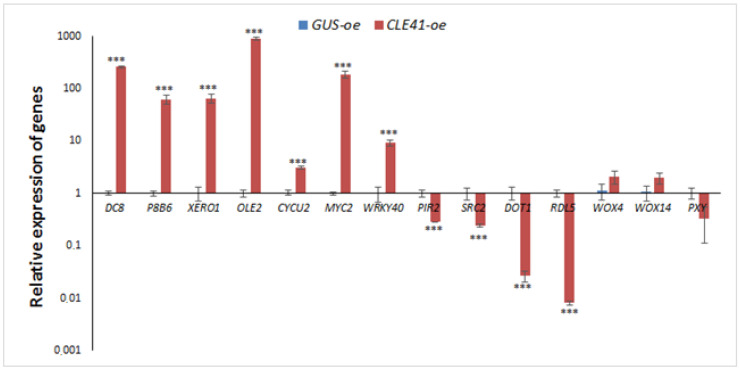
Gene expression analysis by qPCR: bars indicate corresponding genes which were upregulated or downregulated in the transcriptome experiment. Error bars indicate standard deviation of three technical repeats (*** *p* < 0.001).

**Figure 5 plants-11-02163-f005:**
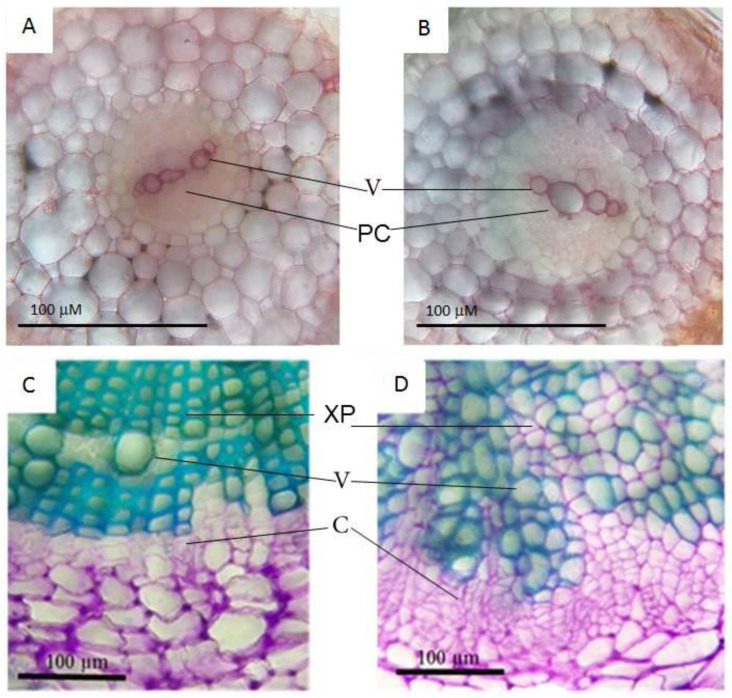
Anatomy of *35S:RsCLE41-1* roots in comparison with *35S:GUS* roots of radish. (**A**)—*35S:RsCLE41-1* roots (7 days old); (**B**)—*35S:GUS 1* roots (7 days old); (**C**)—*35S:RsCLE41-1* roots (30 days old); (**D**)—*35S:GUS 1* roots (30 days old). (**C**,**D**) correspond to Figure 3 in [11]. PC—procambium; C—cambium; V—vessels; XP—xylem parenchyma.

**Figure 6 plants-11-02163-f006:**
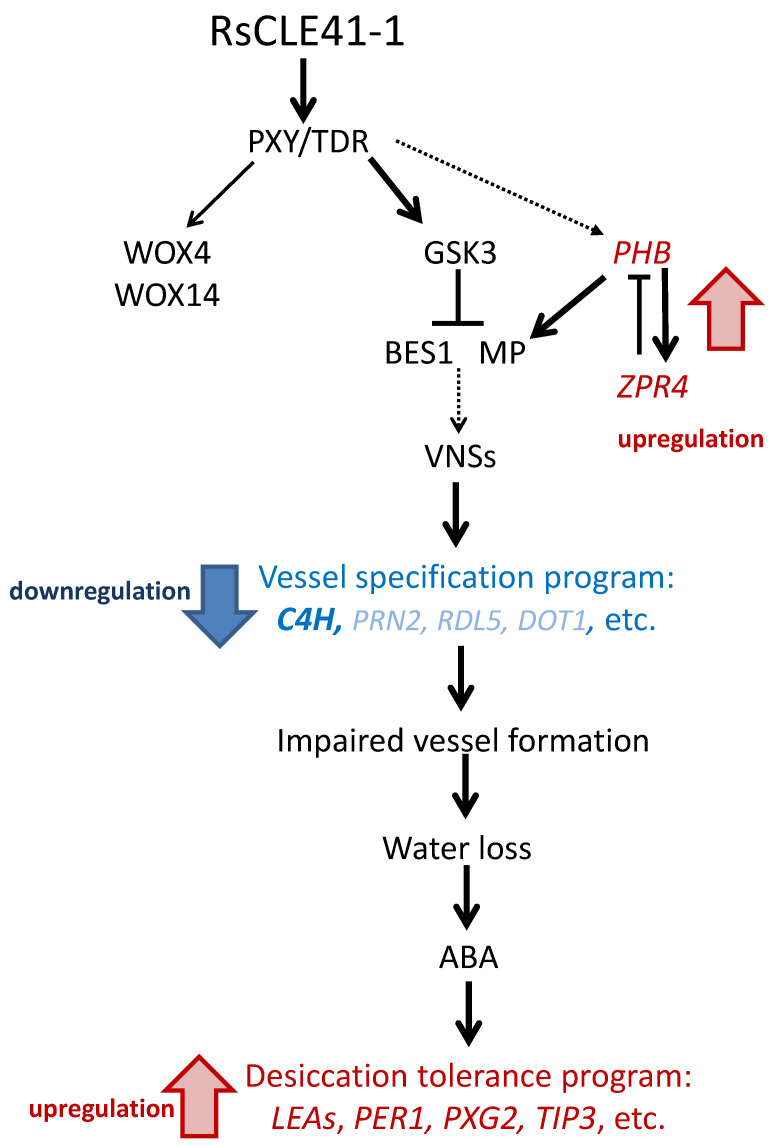
Proposed role of genes involved in vessel specification and ABA-responsive genes which provide desiccation tolerance in the late embryogenesis in the development of *35S:RsCLE41-1* roots.

## Data Availability

Not applicable.

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
