# Peer review of "Transcriptomic Analysis of Radish (Raphanus sativus L.) Roots with CLE41 Overexpression"

_plants, 2022, doi:10.3390/plants11162163_

Round 1

Reviewer 1 Report

The work is dedicated to the analysis of expression levels of genes which were differentially expressed in the 35S:RsCLE41-1 roots in comparison with 35S:GUS roots of radish. In my opinion, the study is very interesting, but the hypothesis requires some additional arguments.

1. Discussion, lines 429-430. RsCLE41-1 overexpression was indicated, but did the authors observe any changes in TDR/PXY? Maybe 7 days wasn't enough for it? The CLE41-PXY signaling is not discussed in the work in any way. Generally, changes in the WOX4, WOX14, etc. expression are preceded by significant changes in the TDR/PXY receptor.

2. Discussion. In my opinion, the anatomical characteristics of the cross-sections should be mentioned in Figure 4.

3. Discussion, lines 464-466. The hypothesis about defects in vascular lignification and xylem specification is questionable. Was there any other evidence of this assumption, besides a decrease in the expression of some genes such as C4H, PRN2, RDL5, DOT1. Any minor changes in the xylem structural elements ratio, as well as in the structure/composition of vessels cell walls, should have affected the change in the expression of the first-layer master switch (VNSs).

Some minor get-up comments are given in the manuscript text.

Kind regards

Author Response

The work is dedicated to the analysis of expression levels of genes which were differentially expressed in the 35S:RsCLE41-1 roots in comparison with 35S:GUS roots of radish. In my opinion, the study is very interesting, but the hypothesis requires some additional arguments.

  1. Discussion, lines 429-430. RsCLE41-1 overexpression was indicated, but did the authors observe any changes in TDR/PXY? Maybe 7 days wasn't enough for it? The CLE41-PXY signaling is not discussed in the work in any way. Generally, changes in the WOX4, WOX14, etc. expression are preceded by significant changes in the TDR/PXY receptor.

We have added a little more information about CLE41-PXY signaling to the Discussion. Also, the qPCR expression analysis of radish PXY, WOX4 and WOX14 genes was added to the subsection 2.4.

  1. In my opinion, the anatomical characteristics of the cross-sections should be mentioned in Figure 4.

We have added the anatomical characteristics of the cross-sections of radish roots to the Figure 4.

  1. Discussion, lines 464-466. The hypothesis about defects in vascular lignification and xylem specification is questionable. Was there any other evidence of this assumption, besides a decrease in the expression of some genes such as C4H, PRN2, RDL5, DOT1. Any minor changes in the xylem structural elements ratio, as well as in the structure/composition of vessels cell walls, should have affected the change in the expression of the first-layer master switch (VNSs).

We have added some information about VNS genes to the Discussion. Unfortunately, the expression of genes from this group did not show significant differences in our transcriptome experiment, although at least the C4H gene may be their target. In this regard, we can hypothesize either the presence of other, VNS-independent pathways for regulating vascular development, or the role of our other down-regulated genes involved in xylem development (such as PRN2, RDL5, DOT1) in these early events associated with changes in vascular development.

  1. Some minor get-up comments are given in the manuscript text.

Thank you, we took into account the reviewer's comments and made the appropriate changes to the manuscript.

Reviewer 2 Report

"Our analysis of transcriptomic data revealed totally 62 differentially expressed genes between transgenic radish roots overexpressing the RsCLE41-1 gene and glucuronidase (GUS) gene" - comparing transgene with GUS is inappropriate. To understand the effect of genes authors need to compare transgenic with empty vector transformed and wild type as well. 

"Among downregulated, genes involved in the stress response were shown to be widely represented" - not clear what the authors mean by "shown to be widely represented"

Figure 3. - Not clear what compared with what. Also better to have the Log2 scale on the y-axis. 

"In total, two samples were analyzed by RNA-seq, representing transgenic roots with CLE41-oe and transgenic roots with GUS-oe as a control (all were in three biological replicates)" - I guess the authors have used three biological samples each for CLE41-oe and GUS-oe genotypes. Better to make this statement very clear. 

Very few genes are up/down regulated therefore making very generalised subheadings like - "Stress-related genes: up- and downregulated" will be misleading 

Authors need to provide pathways or gene networks based on their results instead of descriptive information about each of the differential expressed genes. There should be one figure showing how and exactly which genes get regulated with the overexpression of CLE41. 

At present, the MS is too much descriptive and lacks conclusive observations. 

Author Response

  1. "Our analysis of transcriptomic data revealed totally 62 differentially expressed genes between transgenic radish roots overexpressing the RsCLE41-1 gene and glucuronidase (GUS) gene" - comparing transgene with GUS is inappropriate. To understand the effect of genes authors need to compare transgenic with empty vector transformed and wild type as well. 

Transgenic roots with GUS overexpression can be used as a control for roots with overexpression of CLE41, no worse than an empty vector, since the product of GUS gene, glucuronidase, has no effect on plant development in the absence of a substrate (5-bromo-4-chloro-3-indolyl-beta-D-glucuronide, X-Gluc). Overexpression of the GUS gene is used as a control for studying the effect of overexpression of different genes in various plant species, e.g. https://www.ncbi.nlm.nih.gov/pmc/articles/PMC8125369/ and https://www.cell.com/current-biology/fulltext/S0960-9822(20)30164- https://www.cell.com/action/showPdf?pii=S0960-9822%2820%2930164-0.

  1. "Among downregulated, genes involved in the stress response were shown to be widely represented" - not clear what the authors mean by "shown to be widely represented"

This sentence was rewritten in a clearer manner: “Among downregulated, stress-associated genes were prevailed”.

  1. Figure 3. - Not clear what compared with what. Also better to have the Log2 scale on the y-axis. 

The new Figure 4 (which was previously Figure 3) compares comparative gene expression in roots. We have made a logarithmic scale on the y-axis of this diagram.

  1. Very few genes are up/down regulated therefore making very generalised subheadings like - "Stress-related genes: up- and downregulated" will be misleading. 

Sentence was changed with a clearer wording:”Several genes associated with stress were up- and downregulated in our study”.

  1. Authors need to provide pathways or gene networks based on their results instead of descriptive information about each of the differential expressed genes. There should be one figure showing how and exactly which genes get regulated with the overexpression of CLE41. 

We summarized our obtained results in a gene network (new Figure 3).

  1. At present, the MS is too much descriptive and lacks conclusive observations. 

We have tried to make the Results section less descriptive by shortening the description of gene functions and adding more assumptions.

Round 2

Reviewer 2 Report

No further comments. Revised version looks good.